# Direct ring-strain loading for visible-light accelerated bioorthogonal ligation via diarylsydnone-dibenzo[*b,f*][1,4,5]thiadiazepine photo-click reactions

Jingshuo Gao[1], Qin Xiong[1], Xueting Wu[1], Jiajie Deng[1], Xiaocui Zhang[1], Xiaohu Zhao[1], Pengchi Deng[2✉] & Zhipeng Yu [1✉]

Ultra-fast and selective covalent-bond forming reactions with spatiotemporal controllability are foundational for developing a bioorthogonal approach with high manipulability. However, it is challenging to exploit a reporter functional group to achieve these requirements simultaneously. Here, 11*H*-Dibenzo[*c,f*][1,2]diazepine and a set of heterocyclic analogues are investigated for both their photo-switching natures and their ability to serve as dipolarophiles in photo-click reactions with diarylsydnone. Sulfur-containing dibenzothiadiazepine (DBTD) is discovered to be an excellent chemical reporter in cycloaddition with visible-light excitation for in-situ ring-strain loading via its (*Z*) → (*E*) photo-isomerization. The bioorthogonal utility of the DBTD tag in spatiotemporally controlled ligation for protein modifications on live cells is also demonstrated.

[1] Key Laboratory of Green Chemistry and Technology, Ministry of Education, College of Chemistry, Sichuan University, 29 Wangjiang Road, 610064 Chengdu, China. [2] Analytical and Testing Center, Sichuan University, 29 Wangjiang Road, 610064 Chengdu, China. ✉email: pcdeng@yahoo.com; zhipengy@scu.edu.cn

The burgeoning field of bioorthogonal chemistry[1–3] has played a vital role in biotechnology and biomedical research, as well as providing a unique perspective for studying the native dynamics of biomolecules in real-time. The recent development of "click reactions"[4,5], such as the Huisgen's azide-alkyne cycloaddition[6,7], Staudinger-Bertozzi ligation[7], the inverse-electron-demand Diels-Alder reaction (IEDDA)[8–10] and the photo-click reactions[11], has offered unique tools for chemical ligation that address the rigorous criteria of bioorthogonality. Given its potential to offer superior manipulability in spatial and temporal dimensions[12], photo-click chemistry, in particular, has advanced rapidly over the past two decades[13]. Despite these advances, the development of an effective bioorthogonal photo-click reaction that can provide a precisely-controlled chemical conjugation in live organisms, remains an intriguing challenge. Current progress toward this goal includes various [3 + 2]-cycloadditions, including the photo-decarbonylation of cyclopropenone for azide-alkyne ligation[14,15], nitrile imine (NI)-mediated tetrazole-alkene[16–18], carboxylic acid ligation[19–21] and diarylsydnone (DASyd)-alkene[22–25] photo-click reaction; [4 + 2] transformations, such as visible-light-initiated phenanthrenequinone (PQ)-alkene cycloaddition[26], photocatalytic oxidation of dihydrotetrazine for IEDDA[27], and o-naphthoquinone methides-ene coupling[28]; and thiol-based conjugations, including thiol-ene[29], thiol-yne[30], and thiol-quinone methides[31] additions. In order to achieve ultra-fine photo-control in vivo, it is necessary to introduce a reporter reagent that is bio-inert, able to accelerate the post-photoactivated ligation, and, more importantly, switchable via a harmless self-quenching if the photo-stimulation is withdrawn.

Medium-ring cycloalkenes and cycloalkynes[32,33], which are preloaded with ring-strain so as to elevate cycloaddition reaction rates, are indispensable reagents for bioorthogonal conjugations. Subtle changes in and fine tunings of the ring-strain by either controlling steric influences and/or ring size often leads to appreciable accelerations in reaction rates, for example, 3,3-disubstituted cyclopropene[34] vs. 1,3-disubstituted cyclopropene[35] vs. spiro[2.3]hex-1-ene (sph)[36] for diaryltetrazole photoclick reaction; or trans-cyclooctene (TCO and s-TCO)[37] vs. trans-cycloheptene (TCH and Si-TCH)[38–40] toward s-tetrazine IEDDA. Owing to the inherent instability of those highly strained cycloolefins with respect to thiolysis, hydrolysis and rearrangement, it seems difficult to apply further built-in strain without hampering the practical usage of such ligation reagents in a complex physiological environment. The photo-isomerization of relatively stable (Z)-cycloalkenes to the corresponding highly reactive (E) isomer represents a feasible solution for enhancing the rate of the corresponding bioconjugation in-situ. A notable example of this approach was reported by Weaver (Fig. 1a)[41], in which energy transfer from a photo-excited fac-tris-Ir(4′-CF$_3$-ppy)$_3$ catalyst was used to drive the (E) ⇌ (Z) stationary equilibrium of benzocycloheptene (BC7) so as to favor of the alkene-azide click process. While this report is conceptually important, the photo-energy transfer mediated via the photocatalyst requires two intermolecular interactions to complete the conjugation. Accordingly, there is still an urgent demand to develop direct in-situ ring-strain loadable dipolarophiles, targeting for live cell application.

Dibenzo[b,f][1,4,5]thiadiazepine (DBTD, Fig. 1b) is discovered to be "clickable" with diaryl nitrile imine that is generated via photolysis of DASyd. The resulting reaction yields a macrocyclic azimine imine (MAI, Fig. 1 and Supplementary Table 1). Unlike previous reported dipolarophiles, DBTD possesses an azobenzene core fused into a 7-membered ring structure, allowing for direct ring-strain energy loading in the N=N double bond via a photostationary equilibrium between its (Z) and (E) isomer. The

unique reactivity between the highly ring-strained (E)-azobenzene and nitrile imine encourages us to explore this photo-click reaction as a bioorthogonal chemistry, as reported herein.

## Results and discussion

**Characterization of DASyd-DBDZ photo-click reaction.** Our studies on exploiting the ring-strained azobenzene initiated with the preparation of a series of dibenzo[c,f][1,2]diazepine analogs (DBDZ, Table 1) with various heteroatom fusions via zinc mediated reductive cyclization of the corresponding bis(2-nitrophenyl) compounds (Supplementary Methods)[42]. The DASyds 1d and 1g were chosen as the nitrile imine sources because 1g has an exclusive photo-inducibility under 405 nm excitation whereas 1d does not (Supplementary Fig. 1). This allows us to investigate the DASyd-DBDZ photo-click reaction either via stimulating the photo-conversion of 1d and photo-isomerization process of DBDZ individually, or achieving synergetic photo-activation of 1g in concert with the DBDZ photoisomerization at single 405 nm excitation. To better present the efficiency of the photo-conversion of DASyd to the nitrile imine intermediate, we measured their quantum yield by comparing with a potassium ferrioxalate-based chemical actinometer, showing the $\Phi_{reac.}$ to be 0.154 for 1d and 0.245 for 1g at 311 nm while become undetectable for 1d and 0.081 for 1g under 405 nm laser illumination (Supplementary Fig. 2 and Supplementary Note 1).

To explore the reactivity enhancement via ring-strain energy loading on the DBDZ, comparative analyses of the DBDZ-DASyd photo-click reactions were performed in quartz test tube under irradiation of 311 nm lamp or 405 nm laser (for in-situ isomerization of the DBDZ) or their combination, and the resulting reactions were analyzed by quantitative HPLC-MS (Table 1). The results indicated that: (i) only the sulfur inserted DBTD performed well in the photo-click reaction, giving the stable 3 in excellent yield and suppressing the formation of by-products via hydrolysis and acetonitrile cycloaddition (Table 1, entries 7–12). In contrast, 2a, 2b and 2c without apparently photo-switchable features at 405 nm (Supplementary Fig. 3) showed negligible reactivity despite the complete photo-conversion of DASyd. Hydrolysis or acetonitrile cycloaddition of the reactive nitrile imine intermediate by the solvents were the major side-reactions in the absence of a proper dipolarophile (Table 1, entries 1–6 vs. 7–12, Supplementary Figs. 4–6). (ii) The use of an add-in 405 nm laser led to a significant acceleration with higher yields for cycloaddition with DBTD, presumably due to the photo-induced (Z) → (E) isomerization of its azo moiety rather than assisting the photolysis of 1d (Table 1, entries 7–12, column 8 vs. 10, Supplementary Figs. 7–17); (iii) Intriguingly, the photo-annulation was realized in moderate yields at extremely dilute concentrations (200 nM, Table 1, entries 9 and 12) which would be beneficial for ligation reaction targeting for less abundant biomolecules. (iv) There were no background cycloadditions observed during tracking analyses for up to 116 h if without photoirradiation (Supplementary Fig. 18), thus it provides the foundation to facilitate its spatial-temporal controlled bio-conjugation. In addition, the crystal structure of 3e offers an opportunity to gain insight into the unique twist-boat cyclic structure of the macrocyclic dipole 3 (Table 1), which possesses good stability against nucleophilic additions and dipolarophile attacking (Supplementary Figs. 19 and 20). Under the ambient light conditions for ten days, the decay of the MAI 3 was detected to be less than 6.9% (Supplementary Fig. 21). However, under prolonged irradiation with 311 + 405 nm, the photo-transformation of 3 was observed with a half-life determined to be 36.6 mins (Supplementary Fig. 21). The photo-transformation of 3 resulted in products with the same

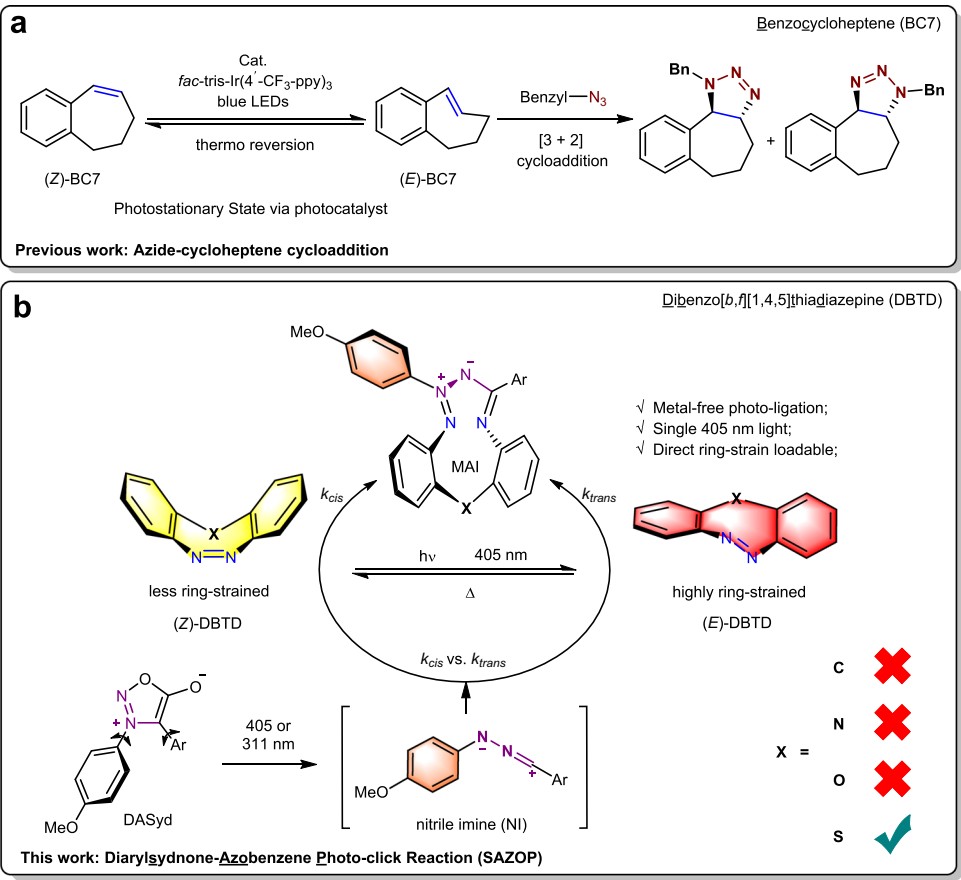

**Fig. 1 Photo-click reactions of seven-membered ring system. a** Indirectly loadable ring-strain in-situ. **b** Directly loadable ring-strain in-situ.

mass-to-charge ratios, suggesting the photo-decaying unlikely led to complete disruption of the covalent-bond linkages between the NI moiety and the DBTD moiety. Interestingly, the macrocyclic adduct **3** could be reduced by glutathione (GSH) (Supplementary Fig. 19d) and tris(2-carboxyethyl)phosphine (TCEP) (Supplementary Fig. 22) instead of forming covalent-bond linkages toward them. It is plausible that, based on the X-ray structure of **3e** (Table 1, Supplementary Table 2 and Supplementary Data 1), the four aryl rings of **3** encapsulate the dipole of the ten-membered MAI from being attacked by slightly large molecules, resulting in a relatively stable dipole. The plausible mechanism of the ring-expansion step of the tetrazolo[1,2-*d*][1,4,5]thiadiazepine intermediate (Supplementary Fig. 23 and Supplementary Note 2) explains the fact that the compound **3** originated from either (*Z*)-DBTD or (*E*)-DBTD converged into the same macrocyclic structure as a favored conformation. Gratifyingly, the DASyd-DBTD photo-click chemistry is worthy of further exploration as a bioorthogonal ligation tool.

**DBTD photo-isomerization property and biocompatibility.**
The DBTD adopts a bowl-shaped conformation in its low-energy (*Z*)-configuration (Fig. 2a, Supplementary Table 3 and Supplementary Data 2), while it is likely to form a twisted-boat structure in the "metastable" (*E*)-isomer which is loaded with a substantial amount of ring-strain energy. There is a scarcity of studies, however, on the photo-physical performance of these photo-switchable cyclo-azobenzenes in relation to the corresponding photo-stationary state (*PSS*). Through an optical-fiber-guided in-situ excitation by a solid-state 405 nm laser during NMR spectra acquisition at 298 K, the conversion of the (*Z*)-DBTD to (*E*)-DBTD reached a *PSS* with a ratio of 5:1 which could be

maintained under continuous illumination (Fig. 2a). When the NMR acquisition temperature decreased to 233 K, the *PSS* ratio of (*Z*)-/(*E*)-DBTD became to be 5:8 (Supplementary Fig. 24 and 25) with eight-fold higher content of the ring-strain loaded (*E*)-isomer. Upon withdrawing the irradiation, the (*E*)-DBTD in the *PSS* was able to rapidly relax to the original (*Z*) state without observation of any decomposition. By utilizing real-time tracking of the absorption evolution at 285 nm, we obtained the first-order kinetic performance of the (*E*)-DBTD to (*Z*)-DBTD thermo-reversions and exhibited the relaxation rate in $3.75 \pm 0.23 \, \text{s}^{-1}$ ($k_{\text{relax}}$) over a course of 3 min in a reciprocating photo-switch process (Fig. 2b and Supplementary Figs. 26 and 27). According to the derivation method[43,44], the photo-switching quantum yield, $\Phi_{\text{ZE405}}$ of (*Z*) → (*E*)-DBTD, was determined to be 0.516 at 298 K (Supplementary Note 3). Unexpectedly, only 9.4% decay of the DBTD after 3.5k times of photo-switching process (Fig. 2c and Supplementary Fig. 28) was detected. Considering the stability in bio-mimicking environment, we also found that the DBTD survived well in the presence of 10 mM GSH either with or without 405 nm laser stimulation (Supplementary Fig. 29). As a result, the excellent photo-fatigue resistance performance of the DBTD makes it as an appropriate dipolarophile for ligation in living systems with fast photo-response. The spatiotemporal controlled photo-isomerization of the DBTD could also be visualized in a transparent glass Dewar at 213 K in ethanol where the decay rate of the (*E*)-DBTD is much slower (Supplementary Fig. 30, please also see in Supplementary Movie 1 as a video material).

**Exploration of photo-acceleration of the photo-click reactions.**
For comparison of the cycloaddition rate of the (*Z*)-DBTD vs.

**Table 1 Photo-activated 1,3-dipolar cycloaddition of DASyd with various DBDZs[a].**

X-ray crystal structure of **3e**

| DBDZ | DASyd | Entry | Conc. of DASyd (μM) | Conversion of DASyd (%) | | | Yield of the desired 3 (%) | | |
|---|---|---|---|---|---|---|---|---|---|
| | | | | 311 nm | 405 nm | 311 + 405 nm | 311 nm | 405 nm | 311 + 405 nm |
| **2a** | **1d** | 1 | 10 | 95 | 2 | 96 | N.D. | N.D. | Trace |
| | **1g** | 2 | 10 | 81 | 65 | 94 | Trace | N.D. | Trace |
| **2b** | **1d** | 3 | 10 | 62 | <1 | 64 | N.D. | N.D. | N.D. |
| | **1g** | 4 | 10 | 79 | 51 | 88 | Trace | N.D. | Trace |
| **2c** | **1d** | 5 | 10 | 80 | 1 | 80 | N.D. | N.D. | Trace |
| | **1g** | 6 | 10 | 70 | 42 | 88 | <5 | <5 | <5 |
| **DBTD** | **1d** | 7 | 10 | >99 | <1 | >99 | 95[b] | <1 | >99 |
| | | 8 | 1.0 | >99[c] | <1[c] | >99[c] | 69[b,c] | N.D.[c] | 84[b,c] |
| | | 9 | 0.2 | 84[d] | <1[d] | 90[d] | 36[d] | N.D.[d] | 57[b,d] |
| | **1g** | 10 | 10 | 86 | 67 | 99 | 82[b] | 63[b] | 85[b] |
| | | 11 | 1.0 | 92[c] | 44[c] | 94[c] | 67[b,c] | 27[b,c] | 79[b,c] |
| | | 12 | 0.2 | 74[e] | 31[e] | 82[e] | 31[b,e] | 12[b,e] | 51[b,e] |

[a]HPLC analysis of photo-induced intermolecular cycloaddition in ACN/H$_2$O (1:1, v/v) under various irradiation condition based on calibration curve. Off-target nitrile imine form photo-conversion of DASyd was primarily quenched by H$_2$O and ACN to form N'-arylbenzohydrazide and 1,3-diaryl-5-methyl-1H-1,2,4-triazole, respectively (Supplementary Figs. 4–6). N.D. = not detected.
[b]The desired product was slightly decomposed under exposure.
[c]for 3 min or
[d]for 2 min or
[e]for 1.5 min.

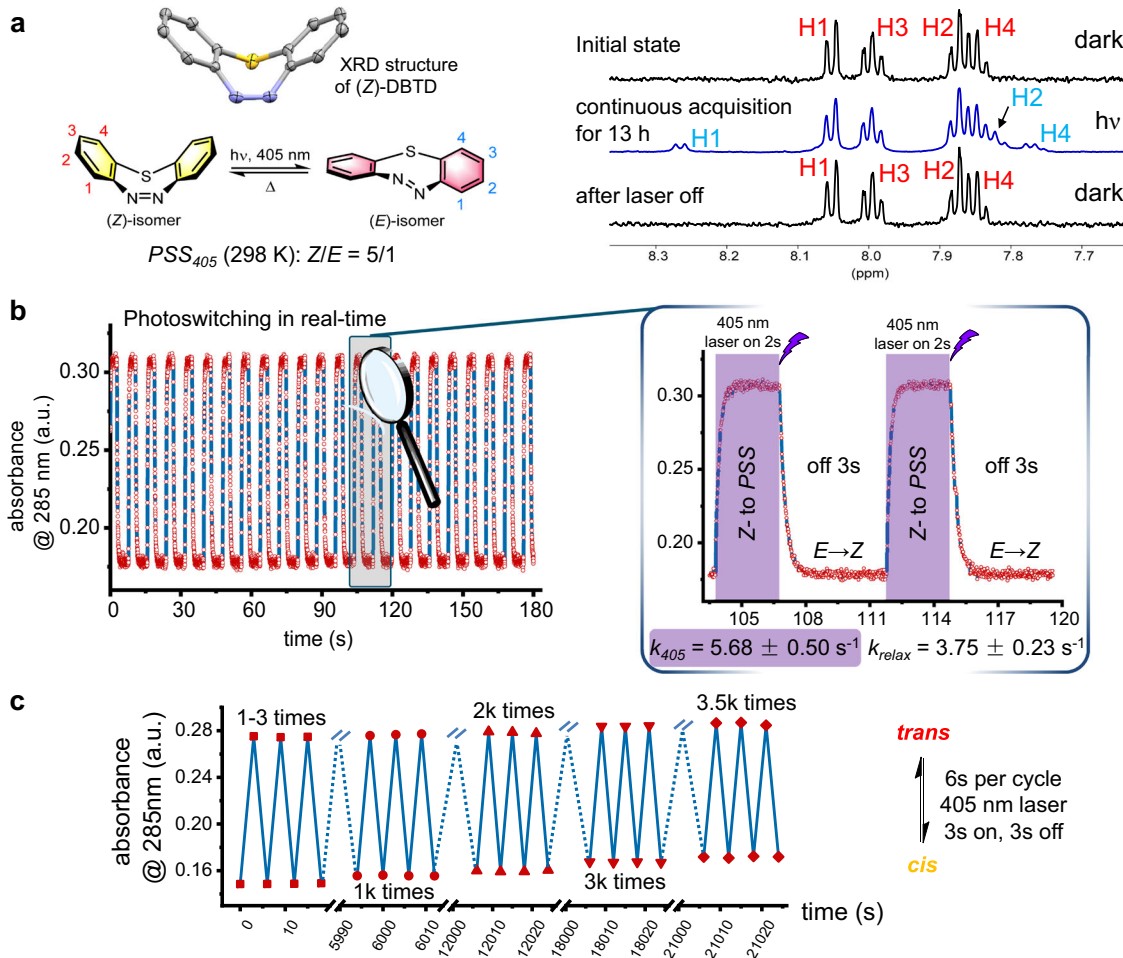

**Fig. 2 Studies of photo-switching properties of the DBTD. a** The isomerization equilibrium and in-situ $^1H$ NMR monitoring (1.0 mM in $D_2O/ACN$-$d_3$ = 1/1, 600 MHz) under 405 nm laser control at 298 K. **b** Real-time photo-switching kinetics between *relax-state* and *PSS-state* with laser off/on, monitored via absorption at 285 nm for 3 min (50 μM). **c** Examination of photo-anti-fatigue performance of DBTD for 5.9 h.

(E)-DBTD toward nitrile imine from photolysis of **1d**, the competitive reaction of DBTD against methacrylamide (DBTD/methacrylamide = 1/3) was recorded by NMR analysis via monitoring the generation of corresponding desired adducts, MAI **3d** and pyrazoline **4d**, to obtain the ratio of their second-order rate constant (Fig. 3a, Supplementary Figs. 31 and 32 and Supplementary Notes 4 and 5). In order to demonstrate the superiority of the in-situ ring-strain loading via photo-isomerization, we set two identical parallel experimental groups which were irradiated with either single 311 nm or 311 + 405 nm combination light, respectively. This strategy was chosen because DASyd **1d** only responses to 311 nm excitation while the DBTD approaches to its *PSS* under 405 nm stimulation. Cycloadduct **3d** and **4d** were found in an 8.3:1 ratio under 311 nm light solely, while the ratio obtained under 311 + 405 nm was 16.1:1. Taking into account that one-sixth of total DBTD was transformed into the (E)-configuration in the *PSS*, the add-on 405 nm laser leads to one-fold increasing in the ratio of **3d/4d** (Fig. 3b). Thus, based on data extraction method (Supplementary Note 5), the (E)-DBTD isomer reacts with a bimolecular rate 6.6-fold faster than that of the (Z)-DBTD (Fig. 3c and Supplementary Fig. 32). To be precise, the relative bimolecular rate constant of the (Z)-DBTD and (E)-DBTD toward nitrile imine was derived as $2.4 \pm 0.24 \times 10^4$ and $1.6 \pm 0.16 \times 10^5$ $M^{-1}s^{-1}$, respectively, whereas that of MAA and the classical TCO were determined to be $0.96 \pm 0.098 \times 10^3$ and $5.0 \pm 0.50 \times 10^3$ $M^{-1}s^{-1}$ (Fig. 3c and Supplementary Fig. 33).

With the aid of 405 nm laser stimulation, the side reactions, including GSH addition and hydrolysis of the nitrile imine intermediate[45], can also be significantly suppressed with the selectivity of the photo-click ligation against GSH improved obviously (Supplementary Figs. 34 and 35). Concretely, we found the ratio between the desire photo-adduct and side-products from GSH + water quenching were increased with 22-fold and 4.5-fold for DASyd **1d** and **1g** under the additional irradiation of the 405 nm laser, respectively. Furthermore, the lyso-DBTD was also photo-ligated with **1d** to form lyso-**3d** in 46% and 91% yield either without or with the acceleration by the 405 nm laser (Fig. 3d and Supplementary Figs. 36–38), respectively. Because carboxylic groups which are widely found in living system could also serve as nucleophiles to quench the NI intermediate[19–21], it is also important to take into account the NI-carboxylic acid ligation when compared with the NI-DBTD cycloaddition. Less than 6.1% of the NI generated from 10 μM DASyd **1d** was trapped by 500 equivalent of Boc-L-Glu-OMe in ACN/PB (phosphate buffer, pH = 7.4) = 1/1 in the presence of 10 equivalent of DBTD under 311 + 405 nm irradiation, the majority of the NI were still able to form the desired MAI **3d** in 86% yield (Supplementary Fig. 39). The LC-MS/MS analysis indicated the specific photo-labeling of the DBTD tag on reside K51 of the lyso-DBTD was illustrated in the peptide fragment profiles of the lysozyme (Supplementary Fig. 40 and Supplementary Note 6). Collectively, the acceleration effect by 405 nm laser activation likely originates from the in-situ

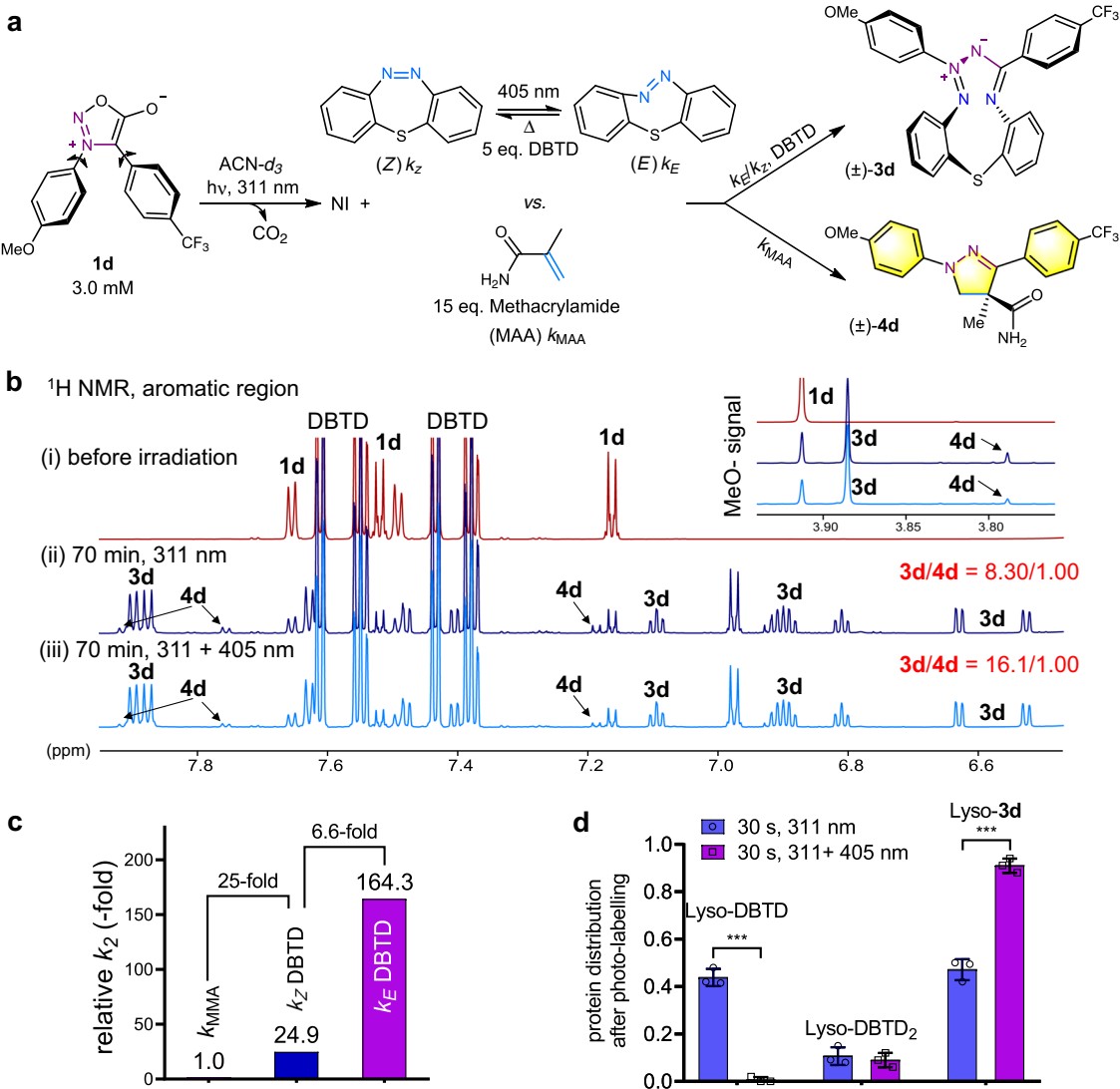

**Fig. 3 Competitive cycloaddition of DBTD versus MAA toward NI. a** Scheme of the competitive reaction model. **b** $^1$H NMR spectra (800 MHz) of the mixture recorded in ACN-$d_3$ (**1d**, $c = 3.0$ mM) at 298 K under various irradiation conditions: (i) before irradiation, (ii) irradiation with 311 nm only and (iii) irradiation with 311 nm + 405 nm laser. **c** Acceleration of the $k_2$ via DBTD isomerization derived from the integrated $^1$H signals (for detailed algorithm, see Supplementary Fig. 30a). **d** Promotion of Lyso-DBTD photo-ligation with 405 nm laser excitation derived from deconvoluted LC-MS spectra after photo-irradiation. Total protein concentration, 2 μM; 50 eq. **1d** in PBS (pH = 7.4). Error bars denote standard deviation from three experimental replicates ($n = 3$); ***$p < 0.001$ from unpaired two-tailed Student's $t$-test.

photo-isomerization of the DBTD and offers higher reactivity and selectivity of the photo-click reaction in complex systems. Therefore, we were able to further utilize the photo-acceleration approach of the DBTD-DASyd cycloaddition to investigate the spatiotemporally controlled bioorthogonal ligation.

**Fluorescent protein labeling via DASyd-DBTD photo-click reactions**. Since both the photo-activation of DASyd **1g**[22–25] and the photo-isomerization of DBTD could be induced by the same 405 nm light source, it is feasible to realize the controlled and accelerated photo-click reaction between DASyd and DBTD under single 405 nm light source to simplify the stimulation procedure and simultaneously reduce the photo-toxicity toward live systems. Intrigued by exploration of single 405 nm induced photo-bioconjugation, we synthesized a **1g-Cy3** conjugate with a fluorescence imaging functionality to study the selectivity toward two DBTD modified proteins (Fig. 4a), lyso-DBTD (lysozyme) and BSA-DBTD/FITC (Albumin from bovine serum). Lyso-DBTD and BSA-DBTD/FITC were subject

to photo-conjugation in cell lysate to mimic live cell environment (Fig. 4a), and then the resultant protein mixtures were analyzed via in-gel fluorescence assay. Via overlapping the FITC (fluorescein-5-isothiocyanate) signal with the Cy3 (Cyanine-3) signal in a multicolor fluorescence imager, the exclusively yellow (labeling of bovine serum albumin, BSA) and red (labeling of lysozyme) bands in the first lane on the SDS-PAGE (Fig. 4b) reflected the high fidelity of the photo-click chemistry for DBTD moieties against the lysate proteins. It is also notable that the selectivity toward the DBTD handles on lysozyme was verified by deconvoluted LC-MS analyses where the control group (native lysozyme) without DBTD residues showed negative results (Fig. 4c, red trace vs. teal trace). Although native lysozyme with several solvent exposed ASP and Glu residues was subject to the optimized photo-click conditions, the carboxylic ligation[19–21] was found to be negligible with **1g-Cy3**, consistent with the small molecule studies in which the selectivity toward the DBTD residues was quite good (Fig. 4c, teal trace). For localized cell imaging purpose, the stability of **3s** in cell lysate was also investigated with

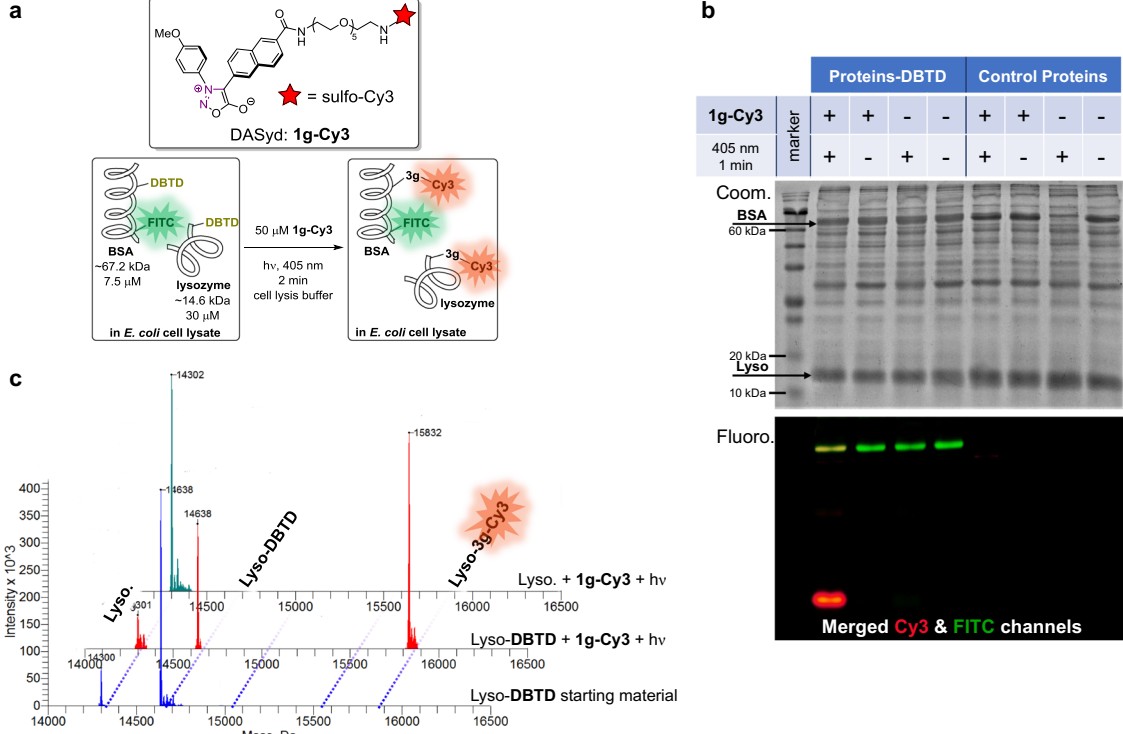

**Fig. 4 Selective fluorescence labeling of proteins-DBTD. a** Schematic illustration of photo-conjugation using single wavelength excitation. **b** Imaging of in-gel fluorescence and Coomassie Blue stained SDS-PAGE. **c** Deconvoluted mass spectra of modification toward lysozyme via DASyd-DBTD photo-click reaction (red) vs. without DBTD modification (teal); total protein concentration 37.5 μM, 50 μM **1g-Cy3** in PBS (pH = 7.4), 405 nm LED array, 2 min.

a decaying half-life of 6.9 h observed (Supplementary Fig. 41). In order to study the influence of MAI toward the linked Cy3-fluorophore, we tracked the fluorescence intensity variation before and after the photo-click reaction which suggested the formation of **3g-Cy3** would partially quench the fluorescence of Cy3, but it could be partially recovered via TCEP reduction (Supplementary Fig. 42).

**Bioorthogonality of DASyd-DBTD photo-click reactions**. To illustrate that the DBTD can mediate visible-light-triggered photo-click ligation in vitro, we treated A549 cells (human pulmonary carcinoma, epidermal growth factor receptor positive, EGFR+)[46] with a bi-conjugated chimeric antibody, Cetuximab (Cex)-DBTD/FITC (Supplementary Fig. 43) to recognize the EGFR-antigen via specific immuno-binding on their surface followed by photo-stimulation in the presence of 7.5 μM **1g-Cy3** (Fig. 5a). Upon irradiation of single 405 nm LED array, the concomitant appearance of the Cy3 and the FITC fluorescence were observed on cell surface via imaging, colocalized precisely, while cells incubated with unmodified Cetuximab exhibited negligible Cy3 signal. (Fig. 5b, upper vs. lower row). Without the photo-irradiation, there was no non-specific binding of the **1g-Cy3** on cell surface detected which suggested the high temporal control of this photo-click chemistry (Fig. 5b, middle row). Interestingly, as shown in Fig. 5c when applying localized illumination, only the 390 nm light exposed cells emitted ring-shaped Cy3 fluorescence pattern, whereas the cells in unexposed portion scarcely has any labeling signal within the microscopic field of view, indicating the high spatial-resolving ability of the DASyd-DBTD photo-ligation in a complex live cell context. After the photo-click ligation with **1g-Cy3**, the A549 cells were lysed and the isolated precipitate were resolved by in-gel assay, which displayed the fluorescence of Cy3 on both the light and heavy chain bands of the Cetuximab (Fig. 5d and Supplementary

Fig. 44) but no fluorescence signal was shown in the SDS-PAGE without the photo-stimulation. Alternatively, the highly regionalized fluorescent labeling of EGFR antigen on A431 cell (human epidermoid carcinoma)[47] membranes were also demonstrated via the **1g-Cy3** photo-ligation toward a Panitumumab-DBTD (Supplementary Fig. 45), a fully human monoclonal antibody conjugate. For the both carcinoma cell lines, the photo-toxicity of the DBTD reporter reagent was evaluated to be minimal at 60 μM dose for 24 h after 30 s exposure to the 405 nm LED illumination (Supplementary Fig. 46).

Collectively, we have developed a visible-light accelerated bioorthogonal ligation tool via [3 + 2] cycloaddition of DBTD toward DASyd mediated through the nitrile imine dipole. The isomerization of DBTD promoted by visible-light rendered the in-situ ring-strain loading on macrocyclic azobenzene to accelerate photo-ligation as a valuable strategy for photo-energy utilization. The exploration of photo-switching performance of the DBTD revealed its fast photo-responsive isomerization with good quantum efficiency and excellent photo-fatigue resistance, which can be controlled for bioorthogonal ligation in a spatiotemporal manner. It is noteworthy that the cycloaddition between the N=N bond of the DBTD toward the nitrile imine dipole to form compound **3** with a twisted macrocyclic structure is ultra-fast, offering a unique reactivity for further investigation. Last but not least, the synthesis of the DBTD is concise compared to those of the highly reactive and ring-strained alkene and alkyne reporters, which represents an easy-access in terms of application prospects.

## Methods
**HPLC analysis**. The photo-induced intermolecular cycloaddition reaction between the DASyds (200 nM, 1 μM or 10 μM) and 5 equiv. DBDZs in ACN/H₂O (1:1, v/v; 0.6% DMSO) under irradiation of the 311 nm, the 405 nm laser or the 311 + 405 nm combined light sources were analyzed by reverse phase HPLC-MS. The stock solution of the DASyds and the DBDZs in DMSO were prepared in the

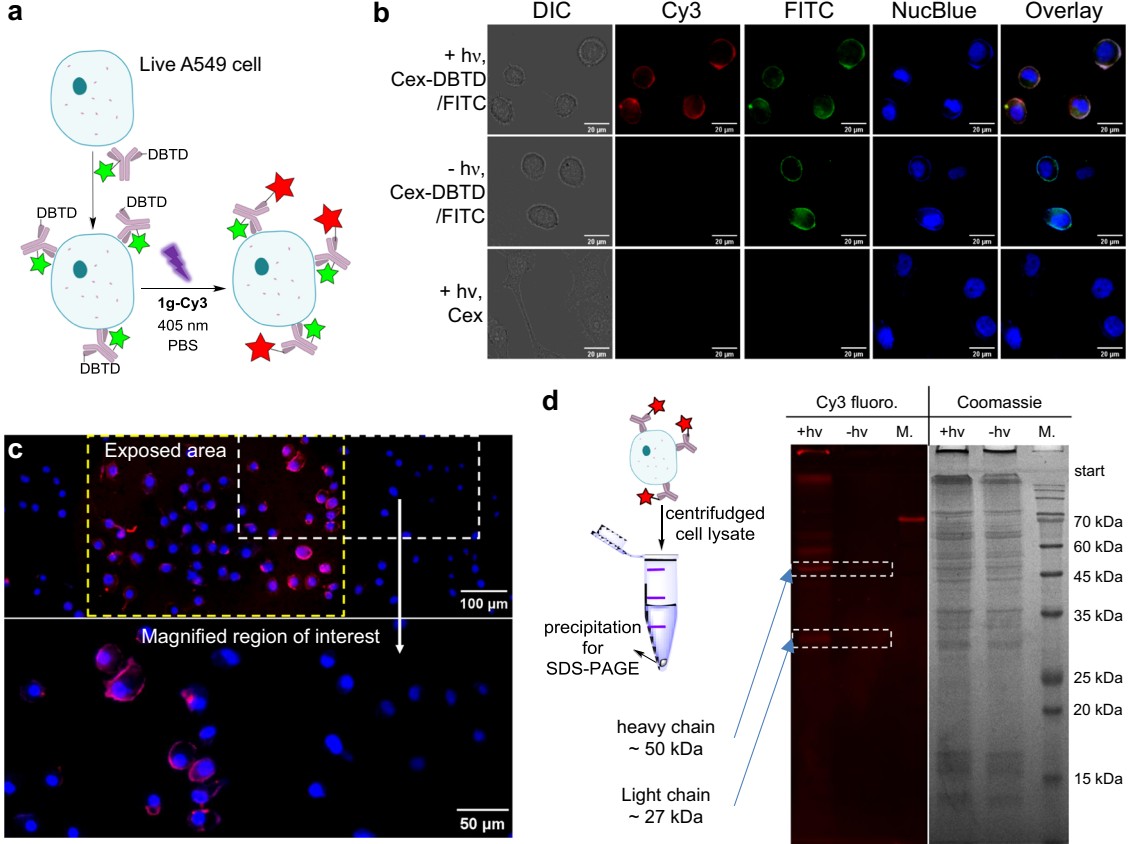

**Fig. 5 Spatiotemporally resolved fluorescence labeling of live cells. a** Schematic illustration of photo-click processes on live cells via 405 nm light induced DASyd-DBTD ligation toward Cetuximab-DBTD (Cex-DBTD). **b** Colocalization imaging experiments of Cyanine-3 (Cy3), fluorescein-5-isothiocyanate (FITC) and NucBlue™ fluorescence channels with control groups. **c** Spatial control of fluorescent labeling on cell membrane surface with irradiation of 390 nm light. **d** SDS-PAGE analysis of the cell lysates of Cex-DBTD bound A549 cells with or without irradiation. M. = protein size marker.

concentration of $10 \times 10^{-3}$ M, respectively. The dual monitoring wavelength of the detector in the HPLC system were 254 nm and 365 nm, the eluents were ACN (0.1% formic acid) and $H_2O$ (0.1% formic acid) at 1 mL min$^{-1}$ flow rate, the gradient of eluents was from ACN/$H_2O$ = 1/9 to 9/1 in a linear format and the stationary phase was a phenomenex Kinetex Carbon18 column (2.6 μ, 100 Å).

**General photo-cycloaddition conditions to isolate the MAIs**. A stirred solution of DASyd (1 mM) and DBTD (1 eq., 1 mM) in 120 mL DCM was irradiated with the 311 nm UV lamp and the 405 nm LED array simultaneously in a quartz round-bottom flask at room temperature for 1.5 h. The solvent was then evaporated, and the residue was purified by silica gel flash chromatography (eluting with 10% EtOAc in hexanes) to give the desired cycloaddition products.

**$^{1}$H, $^{19}$F and $^{13}$C NMR of new compounds**. See Supplementary Figs. 47–106.

**Determination of the *PSS* of DBTD via in-situ NMR monitoring**. The inner tube of a coaxial quartz NMR tube was embedded with a quartz optical-fiber (core diameter: 1000 μm), aligning at the axial direction. The other end of the optical-fiber is combined with an adjustable solid-state 405 nm laser emitter (1W) through the SMA905 interface. The light irradiation was introduced into the interior of the NMR spectrometer by the optical fiber, and the interlayer of the NMR tube containing the sample to be tested can be placed in the sampling chamber of the NMR spectrometer. This setup allows the irradiance and temperature of the sample to be adjusted at any time while acquiring the NMR signals. As the illustration shown (Supplementary Fig. 24), the NMR spectra was obtained through an optical-fiber-guided in-situ excitation of 405 nm laser. The concentration of the substrate in the NMR tube should be as low as possible for even irradiation of light to reach the *PSS*, while maintaining the resonance signal strong enough for better signal/noise ratio within a limited number of acquisition repetitions.

**Photo-switching kinetic analysis of DBTD by 405 nm laser**. The unimolecular rate constants $k_{relax}$ of the photo-switching process of the DBTD was measured via dynamic spectrum tracing upon a 405 nm laser irradiation intermittently in a qpod 2e thermostat cuvette holder (laser on for 2 s and off for 3 s per cycle for display;

laser on for 8 s and off for 8 s per cycle for data analysis and curve fitting) at 50 μM in ACN:$H_2O$ = 1:1 (v/v; 0.1% DMSO). The laser irradiation was sat in front of a quartz lens to adjust the spot size of the laser beam from the laser emitting surface of the solid-state laser to the cuvette, while the cuvette was exposed to the laser irradiation spot as evenly as possible. Mixing appropriate volume of the prepared stock solutions to derive the desired final concentration in sample vials, and the mixture was transferred into a 1.0 × 0.2 cm optical path quartz optical cuvette (405 nm laser irradiation optical path was 0.2 cm). Signals were read out by monitoring the characteristic absorbance signal, indicating the presence of the *trans* state. Kinetic runs were recorded using the following instrumental parameters: monitoring wavelength, $\lambda_{monitor}$ = 285 nm; Interval 17 millisecond per data point over the recorded time range. The data sets were recorded and analyzed with the commercial software, Lightscan. All data processing was performed using Origin pro software.

**Chemical modification of the proteins with the DBTD reporter**. To 0.485 mL solution of protein (lysozyme or BSA, 100 μM in 100 mM NaH$_2$PO$_4$, 25 mM NaOAc, pH = 8.5) was added with DBTD-NHS solution (7.5 μL, 10 mM in DMSO; final concentration = 300 μM). The resulting solution was incubated on a rotating shaker at room temperature for 3 h. Excess amount of small molecules was then removed from the protein mixture by protein spin columns (10 KDa cutoff) using 0.25 M NH$_3$·H$_2$O solution as eluent. And the BSA-DBTD was further modified with FITC (15 μL, 60 eq. 10 mM in DMSO) for 3 h. The final concentration of the protein was quantified with BCA protein assay kit.

**In-gel fluorescence imaging assay**. **1g-Cy3** (500 μM stock solution in water) was mixed with lyso-DBTD (300 μM) and BSA-DBTD/FITC (5 mg mL$^{-1}$) in PBS to derive the final concentration of 50 μM **1g-Cy3**, 30 μM lyso-DBTD and 7.5 μM BSA-DBTD/FITC via dilution with PBS (pH = 7.4), respectively. The mixture was further irradiated with the 405 nm LED array for 30 s and further analyzed by gel electrophoresis using 12% sodium dodecyl sulfate polyacrylamide gels and imaged with a CHAMPCHEMI multicolor fluorescence imaging system using 535 nm (for FITC) and 590 nm (for Cy3) optical filters. Equal protein loading was assessed by staining the same gel with Coomassie Brilliant Blue according to the manual of a commercial staining solution.

**Live cell imaging experiments**. Debita spissitudine A549 cells were seeded in 35-mm glass bottom tissue culture dishes. When reaching over 70% confluency, the A549 cells were incubated with Cetuximab-DBTD/FITC (50 µg mL$^{-1}$) for 40 min and then washed with PBS for three times. A DASyd-fluorophore conjugate, **1g-Cy3** (7.5 µM), was diluted with PBS then added into A549 cells culture dishes, then the whole dishes were irradiated with the 405 nm LED array for 30 s, and further washed with PBS for three times. After the photo-ligation procedure, the NucBlue™ Live ReadyProbes™ (Thermo Fisher Scientific) was added two drops per milliliter and incubated for 15 min at 37 °C to stain the nucleus of the cells. After washing with PBS three times, 2 mL FluoroBrite™ DMEM (Thermo Fisher Scientific) medium was added to the culture dishes before imaging on an Olympus IX83 fluorescence microscopy. After imaging, the A549 cells were lysed with Promega Reporter Lysis Buffer, and the resulting mixture was centrifuged. The supernatant and precipitation of the protein mixture were subject to SDS-PAGE analyses, respectively. The Cetuximab-MAIs from the Lysis Buffer were analyzed via in-gel fluorescence assay.

The spatiotemporal resolved activation of the **1g-Cy3** reagent was realized by using the embedded excitation light source (390 nm optical filter) via an Olympus UPlanSApo ×40 objective on the fluorescence microscope to induce photo-ligation for the spatial controlled imaging. After that, NucBlue™ Live ReadyProbes™ (Thermo Fisher Scientific) was added two drops per milliliter and incubated for 15 min at 37 °C to stain the nucleus of the cells. The A549 cells were then imagined immediately under the fluorescence microscope with corresponding filters in corresponding fluorescence channel.

**Reporting summary**. Further information on research design is available in the Nature Research Reporting Summary linked to this article.

## Data availability
The X-ray crystallographic coordinates for structures, **3e** and (Z)-DBTD, reported in this study have been deposited at the Cambridge Crystallographic Data Centre (CCDC), under deposition numbers 1905280 and 1905279, respectively. Photochemical switching of the $(Z) \rightarrow (E)$ DBTD at 213 K under 405 nm laser stimulation was visualized in the Supplementary Movie 1. The data of these studies are available within the article and its supplementary information files.

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

## Acknowledgements
We thank Dr. Daibing Luo (Analytical and Testing Center, SCU) for help on XRD single crystal analysis and Prof. Jason J. Chruma (SCU) for assistance with paper preparation. Financial support was provided by the National Natural Science Foundation of China (21502130), the "1000-Youth Talents Program", and the Fundamental Research Funds for the Central Universities. We thank the Xiaoming Feng laboratory (SCU) for access to equipments.

## Author contributions
Z.Y. designed the research; J.G. and Z.Y. conceived, and analyzed the experiments; J.G. performed the major experiments; Q.X. and X.W. performed the segmental experiments; J.D. and X.C.Z. provided several samples for experiments; P.D. provided access to resources and support for NMR experimental design; J.G. and Z.Y. co-wrote and revised the paper. X.H.Z. revised the paper.

## Competing interests
The authors declare no competing interests.
