## [Peer review file · Communications Chemistry]

Reviewers' comments:

Reviewer #1 (Remarks to the Author):

In this paper, the author group reports an unprecedented photo-click reaction of a diarylsydnone and a thiadiazepine to produce a macrocyclic azimine imine derivative. They explored the potential of cyclic azobenzene derivatives as a click partner with a nitrile imine that is generated by photolysis of a diarylsydnone, and discovered that a sulfur bridged thiadiazepine efficiently reacts with the nitrile imine. Furthermore, they discovered that a Z-thiadiazepine is isomerized to a metastable E-thiadiazepine by photoexcitation, and the E-isomer is much more reactive than the corresponding Z-isomer. Simultaneous photoexcitation of the diarylsydnone and the thiadiazepine afforded the cycloadduct with high efficiency and selectivity. The reactivity of the E-thiadiazepine toward the nitrile imine was much higher than that of methacrylamide and trans-cyclooctene that have been recognized as click partners of a nitrile imine. The high reactivity of the thiadiazepine contributed to preventing undesired side reaction of the nitrile imine with various molecules such as water, acetonitrile, and GSH. The thiadiazepine and the cycloadduct were stable enough for the use under physiological conditions and the reaction was applied for specific protein labeling in cell lysate and live-cell surface labeling. The location and the timing of the photo-click reaction were completely controllable by the localization and the timing of the light irradiation. Their method realized the rapid and fine conjugation reaction of two components under physiological conditions, and it could be an important option for biorthogonal conjugation reactions.

Overall, this paper contains novel important information including (a) discovery of the nitrile imine-thiadiazepine photo-click chemistry, (b) photochemical property and reactivity of thiadiazepine derivatives, (c) potential of the reaction for the live cell labeling.

The high reactivity and selectivity could be superior to previously reported click chemistry and it could be an important option for biorthogonal conjugation reactions. I will recommend this paper for publication in Communications Chemistry after minor revisions.

Request 1

The author should describe more about the light-stability of the produced azimine-imines.

According to the SI data, the azimine imines absorb visible light. In Table 1, there is a footnote "The desired product was slightly decomposed under exposure." The author should explain more about it. i. e. photodegradation rate or approximate half-life under the reaction conditions and ambient light conditions.

Even if it is difficult to show the detailed photodegradation rate, at least, the author should add a caution about the possibility of the photodegradation.

Request 2

Some groups have reported that a nitrile imine that is generated by photolysis of a tetrazole reacts with a carboxylic acid to form a diacyl hydrazide i. e. (a) Z. Li et al. Angew. Chem. Int. Ed. 2016, 2002, (b) S. Zhao et al. Chem. Commun. 2016 4702. (c) A. Herner et al. J. Am. Chem. Soc. 2016, 14609. The reaction could occur under aqueous conditions and it could be faster than the nitrile imine alkene cycloaddition. The nitrile imine carboxylic acid ligation has been applied to a modification of a native protein via a carboxyl group derived from Asp or Glu residue.

According to the authors' protein labeling experiment, the labeling did not occur without thiadiazepine. Therefore, the nitrile imine carboxylic acid ligation might be negligible under their conditions. However, even if the modification of the carboxyl group did not occur, the author should mention the possibility of the nitrile imine carboxylic acid ligation and add discussion including it.

Request 3

(Z)-/(E)-DBTD ratio at 233 K is 5/8 in the manuscript and 3/5 in SI. One of them should be corrected.

Reviewer #2 (Remarks to the Author):

In this work the authors report a photochemical biorthogonal ligation strategy. There is an established field of cross-disciplinary scientists who are interested in strategies to label molecules

with chemistry like this, thus it is important. This work builds on a strategy previously disclosed by Weaver, in which, photochemical energy is converted into strain energy that is then used to accelerate bimolecular reactions. As the authors point out, the traditional approach of using ever increasing perpetually strained molecules such as the cycloalkynes, to accelerate reactions has a natural limitation, in that at some point the molecules become too strained to handle or react unselectively. This strategy seeks to circumvent this problem by straining molecules in situ and reversibly. The latter is also key to preventing off target reactions from occurring. This reaction differs substantially from Weaver's work in that it is a direct photochemical reaction (actually two), whereas Weaver's was sensitized by an iridium(III) dye. This reaction relies on a photo-release step to "turn on" one partner first, via the irreversible formation of a dipole. The photo-release strategy has distinct advantages but has been explored extensively. The second photochemical step is an isomerization of a cyclic diazo compound to its trans-cyclic diazo which serves to greatly increase the strain energy of this partner. This presumably makes the thermal [3+2] reaction more exothermic and leads to an overall rate enhancement. In order for this to work, the strain-loadable molecule must undergo repetitive isomerization without substantial decomposition as it is anticipated to undergo a thermal relaxation. The authors spend significant effort identifying a diazo partner that can meet these prerequisites. They then go on to show how this can be done to label protein with dyes, and live cells with dyes.

Overall, I think this is an important example that illustrates, in an applied manner, how photochemical energy can be converted into useful strain energy that can be exploited to facilitate the study of chemical biology. It is an attractive and distinct strategy from past efforts which have relied on perpetually strained molecules or purely photo-release strategies. Thus, I am supportive of publication after a few minor issues are resolved.

Table 1. The structure of DASyd should be labeled in the figure component. Currently the reader must match the structure of 1 with the term DASyd.

Table 1. In the structure of the DASyd, the term Ar is used but a phenyl ring is also used. I think just the term Ar is more appropriate, since in some cases it is a naphthyl derivative, which is not a substituted phenyl. This would make the labeling of 1d, e, and g more consistent, since they are already labeled "Ar".

Table 1. Label the DBDZ in the figure as well.

I am concerned about the legibility of many of the schemes. They look like they have been shrunk considerably. Specifically, Fig 2d light grey is bad. 2c the black on blue text is nearly impossible to read. 2a text too small. Fig 3 the MS data is totally illegible. Fig 4 too small text.

P5. "were evaluate" should be "was evaluated".

Upon moving to the studies in figure 3, the authors begin using only the 405 nm light. It is not clear why this change was made and what are the consequences. One can kind of dig this out of the SI, but it was not immediately obvious to me. Will any DASyd work at 405 nm? Or were these chosen specifically for this purpose.

S21. At some point, the authors should acknowledge where the mass balance goes. I presume to solvent or water addition of the DASyd dipole generated upon irradiation.

Point-by-point response to the comments by reviewer #1

- Request 1:** The author should describe more about the light-stability of the produced azimine-imines. According to the SI data, the azimine imines absorb visible light. In Table 1, there is a footnote “The desired product was slightly decomposed under exposure.” The author should explain more about it. i. e. photodegradation rate or approximate half-life under the reaction conditions and ambient light conditions.

Response: We appreciate the reviewer for putting forward this important question. Careful evaluation of the photo-stability of the MAI adduct under photo-irradiation conditions could help us further understand the photochemical properties of these new compounds, and shed light on the application value of this photo-click chemistry. Therefore, the azimine-imine **3** was dissolved in ACN/H₂O = 1/1 and diluted to 100 μM concentration, then HPLC-MS analyses have been carried out to track its decomposition rate. Under ambient light conditions for ten days, the decomposition of the MAI **3** was quite slow which was determined to be less than 6.86% of the initial content. In contrast, significant photo-transformation of MAI **3** was observed with half-life detected to be 36.6 min in the presence of continuous irradiation of both 311 and 405 nm light (Figured S21). The photo-transformation of **3** primarily resulted in several species with the same mass-to-charge ratios, suggesting the photo-decaying unlikely led to complete disruption of the covalent-bond linkages between the NI moiety and the DBTD moiety in the MAI framework. These phenomena suggest that the conjugated structure between the DBTD and NI is most likely intact in those photo-degraded components. Currently, we were unable to accurately sign the chemical structure of those products, but the original purpose of a photo-click conjugation might still be fulfilled even though prolonged exposure was applied. Encouragingly, we are endeavoring to seek a suitable reagent that allows the azimine dipole in compound **3** to be converted into a more stable linkage with fluorogenic properties.

For a better illustration, we have summarized these results and added corresponding descriptions on page 3 of the revised manuscript with highlighted text in yellow background. For details, see the manuscript and SI (Figure S21).

- Request 2:** Some groups have reported that a nitrile imine that is generated by photolysis of a tetrazole reacts with a carboxylic acid to form a diacyl hydrazide i. e. (a) Z. Li et al. *Angew. Chem. Int. Ed.* 2016, 2002, (b) S. Zhao et al. *Chem. Commun.* 2016 4702. (c) A. Herner et al. *J. Am. Chem. Soc.* 2016, 14609. The reaction could occur under aqueous conditions and it could be faster than the nitrile imine alkene cycloaddition. The nitrile imine carboxylic acid ligation has been applied to a modification of a native protein via a carboxyl group derived from Asp or Glu residue.

According to the authors' protein labeling experiment, the labeling did not occur without thiadiazepine. Therefore, the nitrile imine carboxylic acid ligation might be negligible under their conditions. However, even if the modification of the carboxyl group did not occur, the author should mention the possibility of the nitrile imine carboxylic acid ligation and add discussion including it.

Response: The reviewer's suggestions are very instructive. The nitrile imine intermediate generated from photolysis of DASyd is a strong electrophile which could be quenched by GSH (reduced glutathione contains two carboxylic terminals), water and carboxylic acid additions (followed by acryl migration to form diacyl hydrazide). Especially, the tetrazole based carboxylic acid ligation has been developed as a photoaffinity labeling or a protein-crosslinking reagent to discover the drug-target interaction if without the shielding substituents. Although under the conditions of the photo-click reaction between DASyd and DBTD we have explored, there was no protein cross-linking reaction with carboxylic acid residues on lysozyme observed. However, the possibility of a carboxylic acid ligation cannot be ignored. To simplify the study, competing reaction between carboxylic acid and DBTD toward nitrile imine was carried out to further demonstrate the selectivity. Nitrile imine carboxylic acid ligation exhibited an uncompetitive reactivity (see figure S35). Less than 6.1% of the NI generated from 10 μ M DASyd **1d** was trapped by 500 equivalent of Boc-L-Glu-OMe in ACN/PB (phosphate buffer pH = 7.4) in the presence of 10 equivalent of DBTD under 311 + 405 nm irradiation, the majority of the NI were still able to form the desired MAI **3d** in 86% yield (Figure S35).

We are sorry for missing these important references and have made the adjustments to the reference section in our manuscript by adding citations of these research work. The literatures, Angew. Chem. Int. Ed. 2016, 55, 2002–2006, Chem. Commun., 2016, 52, 4702-4705 and J. Am. Chem. Soc. 2016, 138, 14609–14615, exhibit the insightful researches about the photoaffinity-based probes for drug target identification. All of them are quite helpful to enrich introduction information of the manuscript. Therefore, unquoted references have been added to entry 10 of the references. Moreover, we have also made an additional discussion for the possibility of nitrile imine carboxyl acid ligation and highlighted the changes with a yellow background (page 5, left column and page 6 at the bottom of left column).

- 3. Request 3:** (*Z*)-/(*E*)-DBTD ratio at 233 K is 5/8 in the manuscript and 3/5 in SI. One of them should be corrected.

Response: Thanks to the reviewer for pointing out our mistake amicably. We apologize for the careless mistake in the SI, we have changed (*Z*)-/(*E*)-DBTD ratio at 233 K into 5/8 in the SI. To eliminate any potential problem, we have also checked all the data in the manuscript and SI carefully to avoid this kind of mistake. To clearly show the data, we also added the integration areas of the signal peak of both (*Z*)- and (*E*)-DBTD in the ¹H NMR spectra of Figure S24, respectively.

Point-by-point response to the comments by reviewer #2

1. **Request 1:** Table 1. The structure of DASyd should be labeled in the figure component. Currently the reader must match the structure of 1 with the term DASyd.

Response: According to the reviewer's suggestion and we have added the structures of corresponding DASyd in the figure component. Therefore, the reader could directly see the structure with going through the matching. For details see the manuscript, Table 1, column 2.

2. **Request 2:** Table 1. In the structure of the DASyd, the term Ar is used but a phenyl ring is also used. I think just the term Ar is more appropriate, since in some cases it is a naphthyl derivative, which is not a substituted phenyl. This would make the labeling of 1d, e, and g more consistent, since they are already labeled "Ar".

Response: According to your suggestion, we believe simplifying the structure on C-terminal of the DASyd to the abbreviation "Ar" is indeed more appropriate and instructive. DASyd **1g** with naphthyl substituents on its C-terminal was included in our DASyd library. Therefore, the C-terminal structure of the DASyd in both Scheme 1 and Table 1 was redrawn by erasing the phenyl structure.

3. **Request 3:** Table 1. Label the DBDZ in the figure as well.

Response: We have supplied the acronyms (DASyd, DBDZ and MAI) with corresponding compound number below the structures in the figure of Table 1 for clarity.

4. **Request 4:** I am concerned about the legibility of many of the schemes. They look like they have been shrunk considerably. Specifically, Fig 2d light grey is bad. 2c the black on blue text is nearly impossible to read. 2a text too small. Fig 3 the MS data it totally illegible. Fig 4 too small text.

Response: We highly appreciate the reviewer's suggestion. To address the questions about the legibility of many of the schemes, we have edited the typography of original Schemes and Figures from double column style into single column style. The layout in each Scheme and Figures were also adjusted in a clearer format to make it legible to readers. For detailed modifications, the column color of the histogram in Figures 2c and 2d has been revised for better contrast. The black text on the blue column has been changed into white text for better clarity. Since we have expanded the typography of Figure 2, 3 and 4 into single column style, the legibility of the Figures should be much better for readers to see the texts and numbers, especially for the deconvoluted MS spectra in Figure 3c. Hopefully, our revision will offer reader a clear presentation.

5. **Request 5:** P5. "were evaluate" should be "was evaluated".

Response: We apologize for our grammatical error and work hard to improve the English

writing throughout the manuscript.

6. **Request 6:** Upon moving to the studies in figure 3, the authors begin using only the 405 nm light. It is not clear why this change was made and what are the consequences. One can kind of dig this out of the SI, but it was not immediately obvious to me. Will any DASyd work at 405 nm? Or were these chosen specifically for this purpose.

Response: As the reviewer pointed out, the DASyd-DBTD photo-click reaction system requires two photochemical processes on each substrate to complete the accelerated cycloaddition reaction. One of the process is the photo-isomerization of the (*Z*)-DBTD to the “metastable” (*E*)-DBTD, which was induced by 405 nm laser. Another process is the photolysis of DASyd to generate the nitrile imine intermediate. Based on our previous research work in reference 11, we found that the DASyd **1g** with naphthyl substituent on C⁴ position can be activated by 405 nm laser, but other DASyds cannot, e.g. DASyd **1d**. Exclusively, in the case of derivatives from DASyd **1g**, single wavelength light of 405 nm could achieve both the photo-induced conversion of the DASyd into the nitrile imine, and acceleration of the annulation reaction via maintaining the *Z*⇌*E* photo-stationary state of the DBTD. From a practical point of view, we only need to apply single wavelength to trigger the process, but in principle there is an assistance from the *in-situ* isomerization mechanism. It is feasible to realize the controlled and accelerated photo-click reaction between DASyd and DBTD under single 405 nm light source to simplify the stimulation procedure and reduce the photo-toxicity toward live systems simultaneously. As a result, we applied only the 405 nm light in the subsequent study of this bioorthogonal photo-ligation. For clarification, we have also explained the reason for utilization of single 405 nm light at the beginning of first paragraph in the Discussion section to avoid any confusion.

7. **Request 7:** S21. At some point, the authors should acknowledge where the mass balance goes. I presume to solvent or water addition of the DASyd dipole generated upon irradiation.

Response: I would like to express our gratitude to the reviewer for his/her concern about the mass balance of the DASyd-DBTD photo-click reaction. The reviewer’s suggestion is instructive. The nitrile imine formed from photolysis of DASyd is a reactive dipole and can be quenched by solvent molecules, including water addition to form a *N'*-phenylbenzohydrazide (reported in J. AM. CHEM. SOC. 2009, 131, 18036–18037) or acetonitrile addition to form a 1,3-diaryl-5-methyl-1*H*-1,2,4-triazole (reported in J. Org. Chem. 1959, 24, 892-893) or forming a small amount of NI+O and NI+OH adducts which were detected by LC-MS analysis.

We would like to have a detailed explanation herein to the reviewer's concerns about the mass balance in Table S1 (pp 21 in SI). From the experimental data in Table S1, a series of DASyd exhibited more than 95% conversion under the irradiation in the HPLC analysis, but isolated yields of MAI **3** were obtained ranging from 58-81%. There are three reasons giving rise to these results: First, to isolate the MAI **3** for NMR identification of the structure, we had

to carry out the DASyd-DBTD photo-click reactions under much more concentrated conditions (1 mM in dichloromethane) and on a preparative scale (0.12 mmol DASyd, about 40 mg). Secondly, due to the photo-attenuation effect of MAI **3**, the conversion of DASyd was not complete even after 1.5-hour irradiation time, and unconverted starting material was recycled. Under the continuous irradiation of 311 + 405 nm, we also found the desired product of MAI would undergo photo-degradation into several species with the same mass-to-charge ratios (Figure S21), leading to lower isolated yield. Third, the HPLC analyses were performed under conditions: 10 μ M DASyd and 50 μ M DBTD in ACN:H₂O = 1:1, 1 mL volume, 5 min irradiation time. The HPLC-conversions were high because of the diluted concentration.

We have carried out the screening of various azobenzene dipolarophiles (DBDZs) in photo-click with NI from DASyd **1d** (Figure S2-16) and found majority of the by-products were detected as two types (NI+H₂O and NI+ACN) in the LC-MS analysis when there was few desired annulation occurred (Figure S2-4, corresponding peaks were denoted as NI+H₂O and NI+ACN). However, in the presence of the DBTD, the desired cycloaddition reaction became dominant, suppressing the formation of those by-products, thus the LC-MS traces were clean (Figures S7-16, 10 μ M DASyd), indicating the highly efficient transformation of DASyd to MAI with negligible by-products formation except the dihydro-DBTD (Figure S7-16).

This information was supplied in the footnote of Table 1. We have also discussed the presence of water and acetonitrile quenching products in the second paragraph of the Results section which was marked in yellow color. For Table S1 (pp 21 in SI), we have also indicated the reaction scale in head of the column 4 and added the explanation for these moderate isolated yields in the table footnote “b”, and also discussed the presence of water and acetonitrile quenching products in the table footnote “d”. The title of Figure S1 was replaced to avoid any confusion.

REVIEWERS' COMMENTS:

Reviewer #1 (Remarks to the Author):

The manuscript and SI are well revised and this paper is now acceptable for publication in Commun. Chem. without any further correction.

Reviewer #2 (Remarks to the Author):

The authors have carefully and appropriately satisfied the previously raised questions and have, in many places, incorporated these into the manuscript or SI. Therefore, I recommend publication.

Reply to the comments by reviewer 1

We greatly appreciate the reviewers' satisfactory feedback on our revised version.

Reply to the comments by reviewer 2

We wish to express our appreciation for your comments on the revised version.